# Superconducting Quantum Simulation for Many-Body Physics beyond Equilibrium

**DOI:** 10.3390/e26070592

**Published:** 2024-07-11

**Authors:** Yunyan Yao, Liang Xiang

**Affiliations:** ZJU-Hangzhou Global Scientific and Technological Innovation Center, Department of Physics, Zhejiang University, Hangzhou 311200, China

**Keywords:** superconducting quantum simulation, many-body localization, quantum many-body scars, discrete time crystal

## Abstract

Quantum computing is an exciting field that uses quantum principles, such as quantum superposition and entanglement, to tackle complex computational problems. Superconducting quantum circuits, based on Josephson junctions, is one of the most promising physical realizations to achieve the long-term goal of building fault-tolerant quantum computers. The past decade has witnessed the rapid development of this field, where many intermediate-scale multi-qubit experiments emerged to simulate nonequilibrium quantum many-body dynamics that are challenging for classical computers. Here, we review the basic concepts of superconducting quantum simulation and their recent experimental progress in exploring exotic nonequilibrium quantum phenomena emerging in strongly interacting many-body systems, e.g., many-body localization, quantum many-body scars, and discrete time crystals. We further discuss the prospects of quantum simulation experiments to truly solve open problems in nonequilibrium many-body systems.

## 1. Introduction

Quantum computing takes advantage of the intrinsic properties of quantum mechanics, such as quantum entanglement, to efficiently tackle certain tasks like simulating large-scale quantum systems, where classical computing faces exponential resource growth of computation and storage, becoming impractical or infeasible [1,2]. Since the original idea of Richard Feynman [1], quantum computers have attracted extensive attention from the community [3]. Over the past several decades, rapid developments in quantum technologies have forwarded the cutting-edge quantum computing experiments [4,5], from few-qubit systems to the noisy intermediate scale quantum (NISQ) [3].

Nowadays, many quantum platforms showcase the potential for realizing a universal quantum computer in the near future [2,6], such as superconducting qubits [4,7,8], trapped ions [9,10], ultracold atoms [11,12,13], and polarized photons [14,15,16]. Wherein, superconducting qubits hold the advantages of high controllability and scalability [4,6,7] and become the first quantum platform showing quantum supremacy [17,18], indicating the entry of the NISQ era [3]. Current superconducting quantum processors [8,17,18,19] usually exhibit outstanding performances [17,18,19], such as more than 50 programmable qubits, long relaxation times, and high-accuracy control, which are summarized in the table in Section 2. Even though realizing universal quantum computers still demands further improvements in both system size and gate fidelity, these NISQ devices have been successfully used for exploring a wide range of research fields [7,8,20,21,22,23,24], including condensed-matter physics, quantum chemistry, combinatorial optimization, and machine learning.

Many-body physics have attracted extensive attention, such as ergodic dynamics and non-ergodic dynamics. Ergodic dynamics obeys the ergodic hypothesis, which states that a closed quantum system would uniformly explore all possible microscopic states over time [25]. Such ergodic behavior well describes the statistical property of thermalization. Generally, thermalization requires exchanges of energy and particles within different parts of ergodic systems, which leads to the loss of quantum information encoded in the initial state [25]. Whereas, ergodicity-breaking systems can avoid thermalization and have the potential to persist the encoded local information for a long time. Thus, it is of particular interest to explore the ergodicity-breaking systems beyond equilibrium, such as many-body localization (MBL) [25,26], quantum many-body scars (QMBS) [27,28], and Hilbert-space fragmentation [29]. Particularly, such ergodicity-breaking property can be employed to protect Floquet systems from being heating, such as the emerging discrete time crystal (DTC) [30,31,32].

This review aims to summarize the experimental progress in emulating nonequilibrium many-body systems with superconducting quantum circuits. In the presence of local disorders, interacting many-body quantum systems can lead to ergodicity-breaking behaviors that violate equilibrium statistical physics. Here we focus on the many-body localization with strong ergodicity-breaking and quantum many-body scars with weak ergodicity-breaking. As depicted in Figure 1, based on the underlying mechanism of ergodicity breaking, discrete time crystal, a nonequilibrium phase of matter, can be constructed under Floquet drive, where the broken discrete time-translation symmetry gives rise to the subharmonic dynamics. The intermediate-scale quantum processors enable the observation and manipulation of these complex many-body systems, beyond the reach of natural quantum materials. The review is structured as follows. In Section 2, we introduce basic concepts of superconducting quantum simulation. Section 3 focuses on two types of applications for superconducting quantum circuits, exploring nonergodic quantum dynamics and the emerging DTCs. Section 4 provides the summary and outlook based on the development of superconducting quantum computing.

## 2. Superconducting Quantum Simulation

### 2.1. Superconducting Quantum Processors

A superconducting quantum processor generally integrates superconducting qubits, on-chip control lines, readout resonators, and auxiliary couplers [17,33,34,35]. In particular, superconducting qubits comprise lithographically defined Josephson junctions (JJs), capacitors, inductors, and their interconnections.

A JJ usually consists of a very thin insulating barrier clamped by a pair of superconducting electrodes, whose circuit model is equivalent to an anharmonic oscillator combined with a nonlinear inductance ($L_{J}$) and a self-capacitance ($C_{J}$) [36,37,38]. The energy scale of the nonlinear inductance, dubbed Josephson energy, takes the form of $E_{J}=I_{C}\Phi_{0}/2\pi$, with the flux quantum $\Phi_{0}\equiv h/2e$ (*h* is the Planck constant, *e* is the charge of an electron) and critical current of $I_{C}$. The charging energy is $E_{C}={\left(2e\right)}^{2}/2C_{J}$. Providing the ratio $E_{J}/E_{C}\ll 1$, the charge Q=2en (*n* is the number of Cooper pair) stored in the capacitor is a good quantum number, which makes the charging behavior dominant. On the other hand, the Josephson behavior plays a major role if $E_{J}/E_{C}\gg 1$, in which case *Q* is not a good quantum number. Relying on these parameter regimes, charge qubit ($E_{J}/E_{C}\ll 1$), flux qubit ($E_{J}/E_{C}∼10$), and phase qubit ($E_{J}/E_{C}\gg 1$) are developed [7,36,37,39,40]. In addition, several other types of superconducting qubits are under investigation recently based on various mechanisms, such as Andreev-level qubit [41,42], quasi-charge qubit [43], 0−π qubit [44], and so on. These emergent superconducting qubits enrich the foundational building blocks of superconducting quantum computing.

In the past ten years, two-dimensional transmon [45,46,47] qubits have shown great potential in scaling up. The key of a transmon is that the charge qubit is shunted by a large capacitor $C_{s}$ ($C_{s}\gg C_{J}$), which increases the ratio of $E_{J}/E_{C}$ with one order to about 10. This large shunting capacitor makes the transmon less sensitive to the charge noise, and thus a much longer dephasing time is achieved compared with the charge qubit [48,49]. Furthermore, Xmon qubit [50] is a transmon-type qubit with a large capacitor made of a cross shape, which links the microwave driving and flux bias controls, as well as couples to the readout resonator and the nearest-neighbor qubit [50,51]. This artful layout ensures the convenience and efficiency of qubit control, coupling, and readout. At the current stage, most state-of-the-art multi-qubit superconducting quantum processors are built based on transmon qubits, whose performances are summarized in Table 1.

It should be noted that there still are limitations to superconducting quantum circuits. First, challenges toward large-scale superconducting quantum computing. In hardware, cryogenic temperature environment (∼10 mK) and microwave wiring require considerable space volume. A simple expansion of the space volume of a superconducting quantum computer does not seem like a wise choice. Second, noise in quantum circuits. It mainly arises from the control error, the decoherence error due to the short coherence time of qubits, and leakage errors due to the weak anharmonicity of transmon. Therefore, quantum error correction [52] and suppression leakage [53] are urgent for fault-tolerant superconducting quantum computing.

**Table 1 entropy-26-00592-t001:** Some large-scale transmon-based and programmable superconducting quantum processors performances.

Group	Processor (Name, Number)	$T_{1}\;\left(\mu s\right)$	$T_{2}\left(\mu s\right)$	$F_{1q}\;\left(\%\right)$	$F_{2q}\;\left(\%\right)$	$F_{r}\;\left(\%\right)$	Applications	References
Google	*Sycamore*	20	$30^{*}$	99.93	99.5	98.0	QA, QEC,	[17,54,55]
	72				CZ		CMP	
IBM	*ibm_kyiv*	293	157	99.93	99.08	98.47	QA	[56]
	127				CNOT			
Rigetti ^m^	*Ankaa-2*	14.8	8.5	99.75	98.0	95.3	QAOA	[57]
	84				iSWAP			
ETH	*-*	32.5	47.0	99.94	98.8	99.1	QEC	[35]
	17				CZ			
MIT	*-*	22.8	2.4	99.6	-	93.0	CMP	[58]
	16							
USTC	*Zuchongzhi*	30.6	5.3 *	99.91	99.39	95.23	QA	[18]
	66				iSWAP-like			
ZJU ^m^	*Tianmu*	109	17.9 *	99.91	99.4	94.55	CMP	[19,59]
	110				CZ			
IOP CAS	*Chuang-tzu*	21.0	1.2	99.0	-	-	CMP	[60]
	43							
SUST	*-*	25	22.9	99.67	99.0	90.2	CMP	[61]
	16				CZ			

Group: Google: Google AI Quantum, USA; IBM: International Business Machines Quantum Computing, USA; Rigetti: Rigetti Computing, USA; ETH: Eidgenössische Technische Hochschule Zürich, Switzerland; MIT: Massachusetts Institute of Technology, USA; USTC: University of Science and Technology of China, China; ZJU: Zhejiang University, China; IOP CAS: Institute of Physics, Chinese Academy of Sciences, China; SUST: Southern University of Science and Technology, China. Processor: the representative superconducting quantum processor selected from groups, whose performances are summarized from open publications. $T_{1}$, $T_{2}$: The average or median value of the energy relaxation time and dephasing time of controlled qubits, respectively. $F_{1q}$, $F_{2q}$, $F_{r}$: The average or median randomized benchmarking (RB) fidelity of simultaneous single-qubit gate, two-qubit gate (with gate name), and readout, respectively. Applications: applications carried out by the corresponding quantum processor, where QA: quantum advantage; CMP: condensed-matter physics; QEC: quantum error correction; QAOA: quantum approximate optimization algorithm; ^m^: Whose performance data are the median value of statistics, otherwise the average value. *: The result of $T_{2}$ is measured by applying spin echo sequences.

### 2.2. Quantum Logic Gates

Logic operations of a quantum computer can be factorized into a series of single-qubit and two-qubit gates [4]. In general, the Hamiltonian of the qubit control can be expressed as [7]
(1)$$H_{\mathrm{ctl}}=\sum_{i,v}\delta_{i}^{v}\sigma_{i}^{v}+\frac{1}{2}\sum_{ij,v}g_{ij}^{v}\sigma_{i}^{v}\sigma_{j}^{v},$$
where $\sigma^{v}$ is the Pauli operator with v∈{x,y,z}, *i* and *j* label the local qubits. The first term in $H_{\mathrm{ctl}}$ is related to the single-qubit gate operations with a rate of $\delta_{i}^{v}$, and the last term denotes the two-qubit interactions with the coupling of $g^{v}$. Single-qubit operations rotate an arbitrary quantum state into another one on the Bloch sphere. For example, in the computational basis $\{|0\rangle ={\left[1\;0\right]}^{T},|1\rangle ={\left[0\;1\right]}^{T}\}$ of a single qubit, a quantum state |ψ(t)〉 after applying a single-qubit gate $\sigma^{v}$ to initial state $|\psi_{0}\rangle$ can be given as (hereinafter ℏ=1)
(2)$$\begin{array}{c} |\psi \left(t\right)\rangle =e^{-\frac{i}{2}\int_{t_{0}}^{t}\delta^{v}dt^{\prime }\sigma^{v}}|\psi_{0}\rangle =R^{v}\left(\theta \right)|\psi_{0}\rangle , \end{array}$$
where θ is the rotation angle and $R^{v}\left(\theta \right)$ corresponds to the rotation operator. In the Bloch sphere landscape, the matrix representation of single-qubit gates rotating quantum state around the axis of x,y, and *z* is given by
(3)$$\begin{array}{c} R^{v}\left(\theta \right)=\cos\left(\frac{\theta }{2}\right)1-i\sin\left(\frac{\theta }{2}\right)\sigma^{v},\;v\in \left\{x,y,z\right\}. \end{array}$$
When θ=π, the so-called X,Y, and Z gates are formed. Typical single-qubit gates including the identity gate (I), X,Y,Z, and Hadamard gate (H) are listed in Figure 2. Nowadays, implementing a high-fidelity single-qubit gate on superconducting circuits is not a tough task, which can be tuned simultaneously with a fidelity of above 0.999 [19].

Two-qubit entangling gates are essential for universal quantum computation [4]. *i*SWAP gate and controlled-phase (CPHASE) gate are widely employed on superconducting circuits, owing to the tunable interaction between nearest-neighbor qubits. *i*SWAP gate requires the resonant photon swap between states |10〉 and |01〉 (see Figure 3). According to the last term of $H_{\mathrm{ctl}}$, the resonance between two qubits can be represented as
(4)$$H_{2Q}=\frac{g}{2}\left(\sigma_{1}^{x}\sigma_{2}^{x}+\sigma_{1}^{y}\sigma_{2}^{y}\right),$$
which also captures the XY interaction of the system. As a consequence, the unitary operation of such a swap process with a uniform coupling *g* during the evolution time *t* is [62]
(5)$$\begin{array}{ccc} U_{2Q}\left(\theta \right) & = & e^{-i\frac{\theta }{2}\left(\sigma_{1}^{x}\sigma_{2}^{x}+\sigma_{1}^{y}\sigma_{2}^{y}\right)}\\ & = & \left[\begin{array}{cccc} 1 & 0 & 0 & 0\\ 0 & \cos(\theta ) & -i\sin(\theta ) & 0\\ 0 & -i\sin(\theta ) & \cos(\theta ) & 0\\ 0 & 0 & 0 & 1 \end{array}\right], \end{array}$$
with the swap angle θ=gt. An *i*SWAP gate accumulates a swap angle $\theta =\frac{\pi }{2}$ controlled by $t=\frac{\pi }{2g}$. Especially, as $\theta =\frac{\pi }{4}$, $\sqrt{i\mathrm{SWAP}}$ gate is introduced [62], which can be utilized to generate the Bell states [63,64,65,66].

In addition to the resonant swap within the computational subspace, CPHASE gates are implemented by harnessing the avoided crossing between |11〉 and |20〉, which is parameterized by the accumulated conditional phase $\phi =\int_{0}^{\tau }\zeta \left(t\right)dt$, where ζ denotes the frequency deviation between |11〉 and single excitation subspace {|10〉,|01〉} caused by the repulsion effect from |20〉 to |11〉. CZ gate is defined by ϕ=π, which has been demonstrated with a fidelity ≳0.99 in various NISQ superconducting quantum processors [19,54,67,68]. The information of *i*SWAP gate and CZ gate in the computational space are summarized in Figure 3. Leveraging the CPHASE gate, controlled-NOT (CNOT) can be effectuated with the aid of two H gates, expressed as $U_{\mathrm{CNOT}}=\left(1⊗H\right)U_{\mathrm{CZ}}\left(1⊗H\right)$ [62,69].

In terms of benchmarking the fidelity of quantum gates, the generic methods include randomized benchmarking (RB) [70,71,72], cross-entropy benchmarking (XEB) [17,73], compressed sensing [74], and gate set tomography [75]. Here we briefly focus on the first two tools, due to their immunity to the state-preparation and measurement error, and the high efficiency in quantifying and optimizing the performance of quantum gates. The standard RB protocol [76] is used to estimate the average fidelity per gate. RB sequence is composed of randomly sampled Clifford operations and a final inverse operation to keep the qubit initial state unchanged after all these operations. Pauli error ($e_{P}$) of quantum gates can be estimated by fitting the probed fidelity of the final quantum state as a function of the number of gate cycles. XEB [17] also uses random quantum circuits to estimate the gate error of the quantum gate. It requires sampling from a set of random circuits, and calculating the final distribution of bitstrings in all Hilbert space with a classical computer. The fidelity of the quantum gate can be obtained by comparing measured bitstrings with the ideal probability distribution. XEB method not only estimates the fidelity of the quantum gate but also provides the purity of quantum states, which can be used to evaluate the qubit decoherence error. Compared to RB, which is only suitable for Clifford gates, XEB is able to estimate the universal quantum gates with continuous parameters.

### 2.3. Quantum Simulation

Quantum simulation is an important and promising application of superconducting quantum computers. Generally, the time evolution of a quantum system is described by the Schrödinger equation
(6)$$i\frac{\partial }{\partial t}|\psi \left(t\right)\rangle =H|\psi \left(t\right)\rangle ,$$
where *H* is the system Hamiltonian and |ψ(t)〉 is the final state evolved from the initial state $|\psi_{0}\rangle$ (|ψ(t=0)〉) via the unitary time-evolution operator U(t), i.e., $\left|\psi \left(t\right)\rangle =U\left(t\right)\right|\psi_{0}\rangle =e^{-iHt}|\psi_{0}\rangle$. Thus, as shown in Figure 4a, quantum simulation could be realized with three steps: initial state preparation, unitary evolution, and final state measurement. For initial state preparation, a product state on a computational basis is generated by applying single-qubit gates. If the initial state is entangled, additional two-qubit gates have to be applied. It should be emphasized that the easily prepared states usually are the product states.

For unitary evolution, there are two ways to realize it, which are analog quantum simulation (AQS) and digital quantum simulation (DQS) [2]. As for AQS, it directly engineers the time evolution operator U(t) with internal interactions or external fields of the quantum simulators. As depicted in Figure 4b, all qubits are tuned into resonance in the frequency domain, i.e., $\omega_{I}$. The effective coupling between nearest-neighbour qubits is modulated by the frequency detuning (Δ) between resonant qubits and couplers inbetween [77] or the center resonator bus [78]. If considering the on-site frequency disorder, the on-site qubit will be tuned up deviation from $\omega_{I}$. A great advantage of AQS is that it can provide valuable estimations qualitatively even in the presence of errors for some specific problems, such as discriminant MBL, QMBS, and even quantum phase transition.

On the other hand, DQS requires decomposing the time evolution operator U(t) into a series of single- and two-qubit quantum gates, which are described in Section 2.2. Lie–Trotter–Suzuki method [79] is usually used to decompose U(t) into a series of local interactions $\left\{H_{m}\right\}$ that evolve in small time steps Δt, which can be expressed as [2]
(7)$$U\left(\Delta t\right)=e^{-i\sum_{m}H_{m}\Delta t}.$$
As the time step Δt approaches 0, the decomposition results approach the actual time evolution operator U(t). For instance, as shown in Figure 4c, the local interactions $\left\{H_{m}\right\}$ are implemented by a layer of sequentially simultaneous single-qubit *X* and two-qubit CZ gates. Although this scheme is vulnerable to gate errors, in the NISQ era, DQS has shown an outstanding ability to simulate quantum many-body systems with high precision [19,68,80,81]. As introduced below, DQS played a crucial role in simulating the discrete time crystal [32,54,59,67].

Notwithstanding, taking the errors into the numerical simulation of digital quantum circuits is significant for verification. The general error models are the decoherence channel model [82] and the depolarizing channel model [17]. The decoherence channel mainly describes the influence due to energy decay ($T_{1}$) and dephasing ($T_{2}$) of the qubit. The single-qubit decoherence channel is described as
(8)$$\begin{array}{c} E\left(\rho \right)=\left[\begin{array}{cc} 1-\rho_{11}e^{-t/T_{1}} & \rho_{01}e^{-t/T_{2}}\\ \rho_{10}e^{-t/T_{2}} & \rho_{11}e^{-t/T_{1}} \end{array}\right], \end{array}$$
where ρ is the density matrix of single-qubit state with elements $\rho_{\mathrm{ij}}\;\left(i,j\in \left\{0,1\right\}\right)$, and *t* is the effective gate time. It can be simulated by applying reset or $\sigma^{z}$ operations with probabilities related to $t/T_{1}$ and $t/T_{2}$ [82]. As for the depolarizing channel, it can be used to account for the control error. The corresponding depolarizing channel of *d* qubits is
(9)$$\begin{array}{c} G\left(\rho \right)=\left(1-e_{P}\right)\rho +e_{P}/\left(4^{d}-1\right)\sum_{i\neq 0}P_{i}\rho P_{i}, \end{array}$$
where $P_{i}\in {\left\{I,X,Y,Z\right\}}^{⊗d}$ refers to the Pauli string with tensor product of Pauli gates and the Pauli error per gate layer $e_{P}$ is benchmarked by RB or XEB. Numerically, the depolarizing error can be efficiently simulated with the Monte Carlo wavefunction method [83]. For each state-vector evolution, Pauli strings are randomly sampled and inserted into the quantum circuit following each layer of ideal quantum gates with a certain probability.

It must be noted that the hybrid of DQS and AQS is proposed as a digital-analog quantum simulation [84,85,86,87,88,89] by merging digital quantum gates and analog unitary blocks, where the scalability of analog simulation compensates the overhead of a large number of quantum gates.

The state measurement aims to extract the quantum state after the time evolution of U(t). In superconducting quantum circuits, the quantum state should be mapped into the different axes concerning the Bloch sphere before applying the final readout microwave pulses to qubits. Then, the expectation value of the corresponding observable could be obtained. Quantum state tomography (QST) [63,90,91,92] is usually utilized to fully characterize the density matrix of a quantum state. For most problems, there is no need for the whole information of the system and local observables are more efficient, such as the qubit occupation, multi-body correlations, etc. On the other hand, the readout error is an inevitable concern in near-term superconducting quantum processors, arising from the imperfections of state discrimination. Thus, many efforts have been made to mitigate the measurement errors [23,93,94,95,96,97,98,99,100,101,102,103].

## 3. Applications in Many-Body Physics

### 3.1. Quantum Ergodicity and Its Breaking

#### 3.1.1. Quantum Ergodicity

How to understand the reversible microscopic dynamics of an isolated quantum many-body system in the framework of macroscopic statistical mechanics is a fundamental question. In general, an isolated quantum system tends to thermalize itself unaidedly with the memory of initial states lost, which can be explained by the well-known eigenstates thermalization hypothesis (ETH) [104,105,106,107]. It claims that in an ergodic system, individual eigenstates have thermal observables, which are equivalent to the microcanonical ensemble values at certain eigenenergy. Hence, regardless of the entire system being prepared in an eigenstate, its subsystems perceive the remaining eigenstates as an effective heat bath, exploring various configurations solely constrained by global conservation laws. Thus, the ETH of thermalization indicates quantum ergodicity [25].

Level statistics and entanglement structure are two critical properties to benchmark the quantum many-body system, ergodic or not. The former refers to the statistics of the energy spectrum $\left\{E_{m}\right\}$ of the Hamiltonian [108] based on the random matrix theory (RMT) [107]. The distribution (P(r)) of the energy gap ratio $r=min\left\{\Delta E_{m},\Delta E_{m+1}\right\}/max\left\{\Delta E_{m},\Delta E_{m+1}\right\}$ obeys the Wigner–Dyson distribution for ergodic systems [108,109,110], where the level spacing $\Delta E_{m}=E_{m+1}-E_{m}$. Numerically, for the ergodic energy spectrum of a one-dimensional (1D) lattice system, the average value of *r* converges to 〈r〉≅0.53 or 〈r〉≅0.60 in the thermodynamic limit for ensembles with or without time-reversal symmetry [109,110].

As for the entanglement, it can be generically characterized by the entanglement entropy (EE), which aims to describe the microscopic structure of eigenstates and dynamics in quantum systems [111]. EE of a quantum state over the bipartition *A* and *B* can be described as [112]
(10)$$S_{\mathrm{ent}}=-\mathrm{Tr}\left(\rho_{A}\log\rho_{A}\right),$$
where $\rho_{A}$ is the reduced density matrix of subsystem *A* by tracing out subsystem B. For the ergodic system, a volume-law scaling of $S_{\mathrm{ent}}$ is predicted by ETH [25]. That is to say, EE grows proportionally to the volume of subsystem *A*, i.e., $S_{\mathrm{ent}}\propto \mathrm{vol}\left(A\right)$. In the thermodynamic limit, $S_{\mathrm{ent}}$ approaches the Page entropy $S_{\mathrm{Page}}=\left(L_{A}/2\right)\ln2-1/2$ with $L_{A}$ as subsystem size of A, which is the average EE value of entire Hilbert space states.

The rapid progress of quantum hardware enables the experimental observation of ergodic or nonergodic quantum dynamics [113,114,115]. EE in quantum ergodic systems has been observed in superconducting circuits firstly by Neill et al. [113], where only three qubits are involved. The ability to generate arbitrary product states establishes clear signatures of the ergodicity. Chen et al. [115] experimentally study the ergodic dynamics of a 1D chain of 12 superconducting qubits with a transverse field in the hard-core Bose–Hubbard model, and identify the regimes of strong and weak thermalization due to different initial states. A quick saturation of EE to $S_{\mathrm{page}}$ for the strong thermalization is identified with initial states enjoying effective inverse temperature close to 0. These investigations pave the way for further exploring the emergent phenomenon of ergodicity breaking.

For the potential applications in the quantum information process, the ergodicity fails to preserve the quantum information encoded in the initial states, which can be interpreted by the diffusive transport of energy across the whole system within a finite Thouless time [116]. Thus, the existence of the ergodicity breaking triggers many intentions both in theory and experiments, which is expected to memorize the initial conditions for an extremely long time owing to the evading of ETH. Below, we introduce two examples of ergodicity breaking, including MBL and QMBS, which exhibit distinct properties and generation mechanisms.

#### 3.1.2. Many-Body Localization

Many-body localization is an analogy of single-particle localization, namely Anderson Localization (AL) [117], in the many-body framework. Two pivotal elements for realizing MBL are interactions and disorders. Here we start with the 1D MBL model to introduce the basic properties of the MBL system. Then, several experiments for 1D MBL in superconducting quantum circuits [77,78,118] are reviewed. Lastly, we discuss two open issues in this field, including the MBL transition and the dimensionality of MBL.

A representative model for investigating 1D MBL is the spin-1/2 Heisenberg chain with Hamiltonian as [109,116,117]
(11)$$\begin{array}{c} H=\sum_{i}^{L}h_{i}\sigma_{i}^{z}+\sum_{i}^{L}J\left(\sigma_{i}^{x}\sigma_{i+1}^{x}+\sigma_{i}^{y}\sigma_{i+1}^{y}+\Delta \sigma_{i}^{z}\sigma_{i+1}^{z}\right), \end{array}$$
where $h_{i}$ is the random fields at site *i* drawn from a uniform distribution [−h,h], and *J* is the nearest-neighbor interaction, and *L* is the size of the chain. Wherein Δ≠0 leads to the interactions among spins, otherwise Equation (Equation 11) is reduced to the isotropic XY model. As the random field *h* gets larger than the critical field $h_{c}$ and Δ≠0, the MBL phase is formed.

Eigenstates of MBL are completely localized, strongly violating ETH, which indicates the strongly ergocidity-breaking nature of MBL. Eigenstates in the MBL phase inhibit the level repulsion, which supports the Poisson distribution in level statistics with an average adjacent spacing ratio 〈r〉≃0.386, sharply contrasted with the thermal phase. From the perspective of entanglement, MBL leads to an area-law scaling growth of EE [112], i.e., the boundary of the subsystem determining the EE. MBL eigenstates display sparser entanglement structures than that of the ergodic systems, growing logarithmically in time, $S_{\mathrm{ent}}\left(t\right)\propto \ln\left(t\right)$ [112]. For the nonequilibrium dynamics of a MBL system under random disorders, local initial conditions can be preserved, such as particle occupancy $P_{i}\left(t\right)=\left(\langle \sigma_{i}^{z}\left(t\right)\rangle +1\right)/2$ and imbalance $I\left(t\right)=\frac{1}{L}\sum_{i=1}^{L}\langle \sigma_{i}^{z}\left(t\right)\rangle \langle \sigma_{i}^{z}\left(0\right)\rangle$, for an infinite time. In other words, the Thouless energy of the MBL system is much smaller than its level spacing, which leads to insulating transport under bias [119].

The first particular numerical simulation of MBL is carried out by V. Oganesyan and David A. Huse [108], who investigate the level statistics of a 1D strongly interacting spinless fermion chain under random disorders. They pointed out the possible existence of a disorder-induced MBL phase at infinite temperature, opening the door for exploring MBL in theory [25,26,109,120,121,122,123,124,125,126,127,128] and experiments [78,129,130,131,132,133,134,135,136,137,138,139,140,141]. Notwithstanding, it should be noted that the stability of MBL in finite dimensions and thermodynamic limits remains debated actively [124,142,143,144,145,146,147]. Especially as extrapolating the scaling of 1D MBL to h,L→∞ is based on finite-size numerical studies, it is usually misleading [147].

The spectral statistics of the 1D many-body system were first measured by Roushan et al. [77] via the many-body Ramsey spectroscopy technique (MRST). In their work, the Hamiltonian is a Bose–Hubbard model given by
(12)$$\begin{array}{ccc} H_{\mathrm{BH}} & = & \sum_{i}^{L}h_{i}a_{i}^{†}a_{i}+\frac{U}{2}\sum_{i}^{L}n_{i}\left(n_{i}-1\right)+J\sum_{i}^{L-1}\left(a_{i}^{†}a_{i+1}+a_{i+1}^{†}a_{i}\right), \end{array}$$
where $a^{†}\;\left(a\right)$ refers the bosonic creating (annihilation) operator, $n=a_{i}^{†}a_{i}$ is local particle number, $h_{i}$ is the on-site potential, U<0 is the on-site attractive interaction resulted from the anharmonicity, and *J* is the hopping rate of photons between adjacent sites. in superconducting circuits, as |U≫J|, Equation (Equation 12) is called the hard-core Bose–Hubbard model, which can be reduced to an isotropic XY model as described in Equation (Equation 11). The quantum circuit for probing the energy spectrum is shown in Figure 5a. Two qubits ($Q_{m}$ and $Q_{n}$) are initialized in a superposition of |0〉 and |1〉 states by rotation gates with an angle of $\theta =\frac{\pi }{2}$, whereas other qubits remain in the ground state. Then the unitary evolution U(t) of the 1D Bose–Hubbard quantum system is applied via AQS. Before taking projective measurements of $\langle \sigma^{x}\rangle$ or $\langle \sigma^{y}\rangle$, $Q_{m}$ and $Q_{n}$ are applied with the same $\frac{\pi }{2}$-rotation gates as in the state preparation. As consequence, $\chi_{2}\left(m,n\right)=\langle \sigma_{m}^{x}\sigma_{n}^{x}\rangle -\langle \sigma_{m}^{y}\sigma_{n}^{y}\rangle +i\langle \sigma_{m}^{x}\sigma_{n}^{y}\rangle +i\langle \sigma_{m}^{y}\sigma_{n}^{x}\rangle$ is established. The eigenenergies in the two-photon manifold could be obtained from the peaks of the Fourier transform of a series of $\chi_{2}\left(m,n\right)$. Figure 5b exhibits the spectrum statistical results with system disorder strength of 1 and 5, which obey the archetypal Gaussian orthogonal ensemble (GOE), a.k.a., the Wigner–Dyson distribution, and Poisson-type distribution, respectively. These results distinguish MBL from thermalization.

As an aside, the first observation of logarithmic growth of EE $S_{\mathrm{ent}}\left(t\right)$ for the 1D MBL phase was accomplished by Xu et al. [78] in a 10-qubit chain, whose evolution is governed by the engineered spin-1/2 XY model with long-range interactions provided by a central bus resonator. In this experiment, QST is applied to five superconducting qubits to obtain the half-chain EE. In Figure 6, a long-time logarithmic growth of half-chain EE (blue dots) is manifested in the presence of strong disorders. In contrast, for the AL without interactions, EE saturates quickly (green line). Later, many experiments measured the dynamical EE to characterize the MBL transition [58,136,138].

Besides MBL, an unconventional phenomenon of Stark MBL [148] also attracts extensive interest [118,149], which is a many-body version of the single-particle Wannier–Stark localization [150,151]. Stark MBL survives under the linearly tilted potentials, giving rise to the localization in clean systems. Guo et al. [118] experimentally demonstrate the existence of Stark MBL on a 29-qubit superconducting quantum processor, where a linear potential along the device is constructed to emulate the Stark potential. In the experiments, the nonergodic behavior is revealed by probing the two-body correlation function, $C_{i,j}=\left|\langle P_{i}P_{j}\rangle -\langle P_{i}\rangle \langle P_{j}\rangle \right|$, where $P_{i}$ is the local density operator.

Although the crossover from the delocalization regime to the localization regime has been witnessed in various platforms [130,135,152,153,154], the inherent structure of MBL transition is still an open question. The main puzzle is whether the crossover is a phase transition or an intermediate phase that is a hybridization of localization and thermalization [26,109,155]. Quantum critical behaviors of 1D finite-size MBL crossover have been characterized by multi-particle correlations with 12 atoms in an optical lattice under quasiperiodic disorders [135]. The enhanced quantum fluctuation and strong correlation due to the onset of novel diffusive transport were observed in the quantum critical regime.

The existence of a many-body mobility edge at finite energy density is also expected [122], due to the opposite transport behaviors of MBL and thermal systems. Whereas the finite-sized systems make the existence of mobility edge widely debated. In experiments, Guo et al. [137] observed the energy-resolved MBL phase in a 19-qubit superconducting chain with long-range interacting (Figure 7a). The key to observing mobility edge is preparing extensive energy-dependent initial states, which are accessible for superconducting quantum circuits. As shown in Figure 7b, the mapping of steady-state imbalance as functions of energy and disorder strength exhibits mobility edge-like behaviors. This result provides evidence for the existence of mobility edge of 1D MBL transitions. Previously, mobility edge-like behaviors are also investigated by the participation ratio (PR) in a two-photon interacting Bose–Hubbard 9-qubit superconducting chain [77].

Another debated issue is the stability of MBL in higher dimensions [147]. Quantum avalanche theory [124] states that the MBL phase is unstable in systems with the dimension [156] larger than one. An accelerated intrusion of the thermal bath in 1D MBL systems has been observed in a 12-atom chain with one-half of disordered and one-half of the thermal bath [140], which is an overtone of the onset of quantum avalanches in 1D case. The conditions for stabilizing a 1D MBL from stopping the spreading of ergodicity are consistent with the quantum avalanches theory. However, experimental explorations for finite-size 2D MBL provide affirmative signals based on spatial random disorder [131,139,141,157] or quasiperiodic potential [134].

It is worthwhile to note that the Fock-space perspective also provides significant insights for exploring many-body physics. As shown in Figure 8a, the network structure of the Hamming distance (*d*) distributed Fock states illustrates the possible hoppings under the particle-number conservation law, where the initial configuration of $s_{0}$ acts as an apex of such a network. Rather than the experimentally prohibitive inverse participation ratios [128], an experimentally accessible probe is the dynamical radial probability distribution (RPD) [127], which is given by
(13)$$\Pi \left(d,t\right)=\sum_{s\in \{D\left(s,s_{0}\right)=d\}}\left|\langle s\right|e^{-iH_{\mathrm{Fock}}t}|s_{0}{\rangle |}^{2},$$
where $H_{\mathrm{Fock}}$ is the Hamiltonian in Fock space perspective and the state distance $D\left(s,s_{0}\right)=\sum_{i=1}^{L}\left|s_{i}-s_{0,i}\right|$, with $s_{i}\in \left\{0,1\right\}$ standing for the ground state or excited state of the real-space site *i*. Dynamical RPD describes the propagation of the probability wave packet within the Fock space network. The distinct characters of Π(d,t) for three typical many-body states as thermal, critical, and localized states [127] are captured at the bottom of Figure 8a. Yao et al. [141] first observed the 2D Fock space dynamics on a 24-qubit superconducting quantum processor by probing dynamical RPD, where the elusive quantum critical behaviors across the finite-size 2D MBL crossover were indicated quantitatively (Figure 8b). By analyzing the disorder-dependent wave-packet width 〈Δx〉, three distinct regimes were discriminated. The thermal regime (pink shaded area) is identified in disorder below $2J_{0}$ with the same width as the ideal ergodicity ($\Delta x^{\mathrm{Erg}}$). The critical regime (yellow shaded area) is characterized by an anomalous enhancement of 〈Δx〉$≳\Delta x^{\mathrm{Erg}}$ with disorder $2J_{0}≲V≲10J_{0}$, due to the anomalous relaxation behavior [134]. As the disorder increases deeply, the localized regime (blue shaded area) is discriminated with 〈Δx〉<$\Delta x^{\mathrm{Erg}}$ for the nature of localization. The observation of Fock space dynamics is scalable and robust for errors, which provides an efficient approach to further exploring the controversial issues in 2D many-body system [127,158], such as the delocalization and mobility edge in 2D disordered many-body systems.

#### 3.1.3. Quantum Many-Body Scars

Besides the strong breakdown of ergodicity like MBL, an intriguing weak ergodicity breaking state in interacting many-body systems is quantum many-body scars, where only a small set of eigenstates at finite energy density evade ETH, while most eigenstates still obey the thermalization fate. QMBS are a many-body analogy of quantum scar in the single-particle motions in quantum billiards [28] along the periodic trajectory that is unstable. Without loss of generality, inspired by the Rydberg atoms [159], the Hamiltonian of PXP model is given as [27]
(14)$$\begin{array}{c} H_{\mathrm{PXP}}=\sum_{i=1}^{L}P_{i-1}\sigma_{i}^{x}P_{i+1}, \end{array}$$
where projector $P_{i}=\left(1-\sigma_{i}^{z}\right)/2$ ensures no simultaneous excitations of the adjacent atoms. The Hilbert space of the PXP model is constrained, whose dimension is equal to $D=F_{L-1}+F_{L+1}$ ($F_{L}$ is the *L*-th Fibonacci number) for the periodic boundary conditions. Its energy spectrum embodies spaced towers, which are unveiled by the anomalously enhanced overlap of special eigenstates with others at the same energy density. Such special eigenstates within the middle of the spectrum are dubbed QMBS, which are formed in a decoupled subregion within the Hilbert space. Therefore, initial states such as QMBS showcase perfect revivals with a single frequency during unitary evolution if the towers are equidistant.

QMBS are predicted to obey the Wigner–Dyson distribution of energy level statistics akin to the thermal states [27,160,161], indicating a distinct level of repulsion and chaotic behaviors. This is in stark contrast with integrable and disordered systems. QMBS are the low-entanglement eigenstates, where EE grows with system size governed by the volume law [29,162]. As for the quenched dynamics as initialized with QMBS, EE grows linearly in time accompanied by weak oscillations [27,163]. Overall, QMBS is expected to own robustness to perturbations and more generous entanglement structures than MBL, which can be employed for the quantum information process, preparation, and manipulation of entangled states [59,164], and especially in quantum metrology [165].

Otherwise the kinetically constrained PXP model of QMBS [159,166,167], many other models with QMBS are implemented experimentally, such as the tilted Bose–Hubbard model in optical lattices [166], and anisotropic Heisenberg magnets [168]. A distinct type of QMBS embedded in non-constrained kinetic models was realized on superconducting quantum processors [169,170,171]. Zhang et al. [169] generated a new type of QMBS by weakly decoupling one part of the Hilbert space in the computational basis. In this experiment, the quasi-1D hard-core Bose–Hubbard model is effectively emulated with the approach of AQS, whose integrability is broken by the parasitic random longer-range cross-couplings. In detail, the periodic coherent dynamics of fidelity and subsystem 4-qubit entanglement entropy in a 30-qubit chain of QMBS can be found in Figure 9. Compared to the thermal initial state, EE of QMBS exhibits a slower growth. Furthermore, Dong et al. [170] demonstrated the modulation of exact QMBS with rich entanglement structure in a 16-qubit ladder superconducting quantum circuits via engineering inhomogeneous spatial couplings. These works extend the universality of QMBS in the regime of unconstrained dynamics.

On the other hand, via DQS, Chen et al. [171] utilized up to 19 qubits on IBM quantum processors to probe the nontrivial dynamics of QMBS in an Ising model with transverse and longitudinal fields, where the scaled cross-resonance gates and various quantum error-mitigation techniques are employed. The observed coherent oscillations of the connected unequal-time correlation function $C_{Y}\left(t\right)$ demonstrated the essential coherence and long-range many-body correlations of the QMBS eigenstates with $Z_{2}$ symmetry.

### 3.2. Ergodicity-Breaking Protected Discrete Time Crystal

It is said that where ergodicity breaks, there will be a discrete time crystal [32]. This is due to that the ergodicity-breaking states, such as MBL and QMBS, can be exploited to avoid heating under Floquet drive [25]. Here, we introduce the applications of MBL and QMBS in protecting the nonequilibrium phase of matter, the discrete time crystal. Before that, we would like to introduce the concept of the discrete time crystal.

Time crystals have a rich historical background, characterized by the phases of matter that spontaneously break the time translation symmetry (TTS) [172]. In 2012, Frank Wilczek [173] proposed a concrete conformation of continuous quantum time crystal in macroscopic quantum systems. In this toy model, the ground state periodically breaks the time translation symmetry, analogous to crystals in space which breaks the spatial translation symmetry. However, the realization of such a time crystal is deemed implausible within thermal equilibrium frameworks and local static Hamiltonians [174,175,176], where late-time equilibrium states remain time-invariant. Fortunately, non-equilibrium systems, particularly Floquet systems, offer the promise to realize time crystals.

The Hamiltonian of Floquet systems exhibits discrete TTS, in which the time translation is a subharmonic behavior, to wit, $H\left(t+nT\right)=H\left(t\right),1<n\in N^{+}$ [177]. In the concept of Floquet time crystal in quantum many-body systems [177,178,179], spontaneous breaking of discrete TTS can be characterized by the subharmonic response of certain observables, such as local magnetization [180], which is robust to generic perturbations and endures infinitely as approaching the thermodynamic limit. Such Floquet time crystal is also dubbed π spin glass [177,179], which is predicted to exhibit spatial long-range correlations, a new type of spatiotemporal order. However, Floquet driving induces additional energy injection to heat the isolated quantum many-body system to an infinite temperature state, which is incompatible with the spatiotemporal order. For practical purposes and future applications, it is proposed to exploit ergodicity breaking to mitigate the risk of heating under perturbations [177,178,179].

MBL is a natural candidate for protecting DTC with spatial disorders under Floquet driving. In a 1D spin-1/2 system, the corresponding Floquet driving Hamiltonian $H_{F}$ is given as [178]
$$\begin{array}{ccc} H_{F} & = & \left\{\begin{array}{cc} H_{\mathrm{MBL}}=\sum_{i}\left(J_{i}\sigma_{i}^{z}\sigma_{i+1}^{z}+h_{i}\sigma_{i}^{z}+\delta_{i}\sigma_{i}^{x}\right), & 0<t<t_{0},\\ H_{\mathrm{Flip}}=g\sum_{i}\sigma_{i}^{x}, & t_{0}<t<t_{0}+t_{1}, \end{array}\right. \end{array}$$
where the $H_{\mathrm{MBL}}$ guarantees the evading ETH of eigenstates, and $H_{\mathrm{Flip}}$ flips spins with certain time $t_{1}$. Generally, parameters $J_{i}$, $h_{i}$, and $\delta_{i}$ in $H_{\mathrm{MBL}}$ are chosen from certain random distributions, and the driving period is $T=t_{0}+t_{1}$. For simplicity, $\delta_{i}$ keeps zero such that the eigenstates of $H_{\mathrm{MBL}}$ are eigenstates of the individual $\sigma_{i}^{z}$, which conserves the spin-flip operation $e^{-i\frac{\pi }{2}\sum_{i}\sigma_{i}^{x}}=\prod_{i}\sigma_{i}^{x}$ as $t_{1}=\pi /2g$. For any individual spin initialized along the *z* direction, the spin-flip operation acts as a π pulse for each spin, which leads to a period-doubled response of local observables. In this case, the disordered Floquet system is robust to small perturbations, and the subharmonic response is rigid even as the initial states and Hamiltonian perturbated weakly.

Besides the MBL, a range of mechanisms have been investigated theoretically and experimentally for implementing genuine DTC, which go from prethermalization [82,181,182,183,184,185,186] to weak ergodicity breaking [187,188,189,190,191,192,193,194] and interacting integrability [195], as well as ancillary protecting symmetries [196,197,198]. Below, we introduce the many-body localized discrete time crystal (MBL-DTC) [54,67,178,199,200,201,202] and quantum many-body scarred discrete time crystal (SDTC) [32,59,187,193,194] in the strongly interacting many-body systems, respectively.

#### 3.2.1. Many-Body Localized Discrete Time Crystal

The hallmark of MBL-DTC is the oscillation of local observables at a period-doubling of Floquet drive [32], exhibiting spatiotemporal long-range order regardless of generic initial states. The lifetime of MBL-DTC is predicted to be infinite for fully localized eigenstates in Hilbert space. Ippoloti et al. [180] pointed out that the genuine MBL-DTC can be implemented in superconducting quantum processors with quantum digital circuits. Compared with DTC signatures implemented in other quantum platforms, such as trapped ions [203], NV centers [204], and NMR spin [205], superconducting circuits can fully satisfy the requirements for stabilizing a genuine MBL-DTC [180], which includes the Ising-even disorder, short-range interactions, various initial states and site-resolved measurement. Notwithstanding this, the MBL-DTC also needs to face a similar dilemma as the MBL does.

Subsequently, the MBL-DTC was demonstrated by Mi et al. [54] with a 20-qubit chain isolated on a Google Sycamore processor through digital quantum circuits, where the average Pauli error per cycle consisting of two single-qubit gates and a tunable two-qubit CPHASE gate is 0.011. As depicted in Figure 10a, the Hamiltonian is
(15)$$\begin{array}{c} H_{F}=\frac{1}{2}\sum_{i}h_{i}\sigma_{i}^{z}+\frac{1}{4}\sum_{i}\Phi_{i}\sigma_{i}^{z}\sigma_{i+1}^{z}+\frac{\pi g}{2}\sum_{i}\sigma_{i}^{x}, \end{array}$$
corresponding to the longitudinal fields, nearest-neighbor Ising interactions, and imperfect global spin flips, respectively. The strong and disordered Ising-even interactions $\Phi_{i}\sigma_{i}^{z}\sigma_{i+1}^{z}$ guarantee the long-range spatial order essential for the robustness of DTC [177,180]. In addition, the last term of $H_{F}$ controls the qubits flipping by a parameter of *g*, which can be tuned to witness the phase transition from DTC to thermalization with a critical value $g_{c}=0.84$ [180]. The long-lived oscillations of disorder and initial-state averaged autocorrelator $\bar{A}=\bar{\langle Z(0)Z(t)\rangle }$ in MBL-DTC with a value of g=0.97 (Figure 10b, right panel), which is sharply contrasted with a thermal phase of g=0.60 with fast dissipation (Figure 10b, left panel). It notes that the subharmonic oscillation persists up to 100 cycles, whose total operation time is comparable with the coherence time of quantum circuits. The impact of slow internal thermalization on the dissipative oscillations in MBL-DTC was ruled out by time-reversal techniques [205].

On the other hand, Frey et al. [202] observed the signatures of MBL-DTC via the AQS strategy on a 57-qubit superconducting chain isolated from quantum processors ibmq_manhattan and ibmq_brooklyn of IBM. Meanwhile, 1D Floquet symmetry-protected topological phases (FSPT), a.k.a., symmetry-protected time crystal, was implemented on a 26-qubit superconducting chain by Zhang et al. [67], where the discrete time-translational symmetry only breaks at the boundaries rather than the bulk. In reality, the FSPT phase is essentially protected by the MBL engineered, no matter the boundaries or the bulk.

#### 3.2.2. Quantum Many-Body Scarred Discrete Time Crystal

Unlike in the MBL-DTC where all eigenstates are localized, in the quantum many-body scarred DTC [188,189,190,191,192,193], only certain scarred subregions of eigenstates are decoupled from the ergodic Hilbert space. An atypical and toy model of SDTC based on a 1D chain with periodic boundaries is [193]
(16)$$\begin{array}{ccc} U_{F} & = & e^{-i\theta N}e^{-i\tau H_{\mathrm{PXP}}}, \end{array}$$
where $N=\sum_{i}n_{i}$ is the particle number operator controlled by rotation angle θ, and $H_{\mathrm{PXP}}=\sum_{i}P_{i}\sigma_{i+1}^{x}P_{i+2}$ is the PXP Hamiltonian for time τ as described above in Equation (Equation 14). As θ=0 in Equation (Equation 16), the evolution is governed by $H_{\mathrm{PXP}}$, whose scarred states are the pair of Néel-like state $|Z_{2}\rangle =|1010\cdots \rangle$ and its inversion.

It is crucial to properly initialize the Floquet systems within the subregion of QMBS. The hopping within the subregion gives rise to the long-lasting stroboscopic dynamics of SDTC. However, due to the final hybridization of the scarred and thermal eigenstates, SDTC persists partially and features rapid growth of EE [193]. For the strong driving case with a centerboard range at θ≈π and $\tau =T/2=\pi /\omega_{D}$, the Néel-like states exhibit perfect and long-lasting oscillations, where the spreading of entanglement is extremely suppressed [193]. In contrast, for the initial states far away from scarring states, PXP evolution leads to ergodic spreading. On the other hand, SDTC expresses some distinctive performances beyond MBL-DTC. Due to the inherent quantum chaotic nature of SDTC, it provides a platform to explore robust time-crystalline behaviors even in the presence of quantum chaos [188], which is an open problem at present. Moreover, there are no concerns about dimensionality or disorder strengths to effectuate an SDTC.

The programmable Rydberg atom quantum simulators [187] have observed the signatures of the SDTC, where the Floquet systems range from chains and square lattices to exotic decorated lattices with natural long-range interactions. The Floquet evolution is based on the Bose–Hubbard model. The emergent subharmonic locking and stabilization are unveiled by a long-lived oscillatory imbalance and slow growth of EE, under the different regimes of geometries and system size. This work demonstrates the ability to steer entanglement dynamics in complex systems with the combination of scar states and Floquet driving.

Furthermore, recently, the cat scar DTC was analyzed [194] with the engineering of several pairs of cat eigenstates with strong Ising interactions, which is robust against generic perturbations and presents oscillatory dynamics for an exponentially growing time $∼e^{L}$. The pair number of cat states can be controlled by both the patterns of states and the number of various interaction strengths. These scars hold a pairwise rigid quasienergy difference of π, indicating the pairwise localizations in Fock space. Hence, the cat scars are believed to hold long-range correlations.

Experimentally, in 2D grid superconducting chip, Bao et al. [59] achieved cat scar enforced DTC, which is exploited to prolong the lifetime for a generic Greenberger–Horne–Zeilinger (GHZ) state. As depicted in Figure 11a, the created GHZ state is embedded in a cat scar DTC, together with a reverse circuit to construct Schrödinger cat interferometry. Using such interferometry, one can simultaneously benchmark the macroscopic coherence of relative phase in cat scars, and the temporal structure leading to the period-doubled oscillation of such macroscopic phases. The digital DTC unitary $U_{F}$ consists of two layers of two-qubit CZ gates as Ising interactions and two layers of single-qubit gates, where the averaged simultaneous fidelity of single-qubit gates and two-qubit CZ gates are 0.999 and 0.995, respectively. A qualitative distinction in Figure 11b indicates that the cat scar DTC persists in the stroboscopic oscillation of relative coherence phases in a GHZ state throughout 30 cycles, producing a robust quantum state of matter to protect GHZ states from perturbations. In this experiment, the deepest quantum circuits have an operation time of more than 15 μs. These results provide a novel strategy for protecting large-scale GHZ states with a nonequilibrium phase of matter.

## 4. Conclusions and Outlook

This review briefly summarizes the applications of superconducting quantum simulation in many-body physics. In particular, we highlight several recent experimental progresses in the realm of quantum ergodicity and two aspects of ergodicity breaking: MBL and QMBS. For 1D MBL, the Poisson distribution of level statistics, logarithmic growth of EE, mobility edges, and phase transitions are observed experimentally. In addition, Fock space also sheds light on controversial issues, i.e., the dimensionality of stable MBL. The kinetic-unconstrained QMBS was demonstrated in the superconducting qubits chain. Meanwhile, the novel out-of-equilibrium phase of matter, discrete time crystal, is implemented deterministically with the digital quantum circuits under the protection of MBL and QMBS. Despite the existence of noise in NISQ devices, superconducting circuits still exhibit remarkable potential in advancing the study of many-body physics.

We will continue optimizing the performance of superconducting quantum circuits, by prolonging the coherence time of qubits and reducing crosstalk within qubits. The advanced superconducting quantum circuits would play a vital role in understanding the nature of many-body systems at large scale and long time regimes. Here, we outline several directions for possible explorations in the future. First, there are open problems in many-body physics, such as MBL transitions, the dimensionality of stable MBL, and even the possibility of disorder-free MBL, etc. The key challenge for effective scaling of MBL transitions is the remarkable finite-size effect. It can be tackled by taking quantum simulations of larger system sizes with increasing the number of qubits. Besides, it is beneficial to find proper observables that exhibit less severe finite-size effects. Fock space may provide a new insight into this hardness [158]. The spectral form factor may be an accessible observable [206] to extract Thouless energy, which is an alternative diagnosis of thermalization and localization [145,147]. Second, demonstrating more counterexamples of quantum ergodicity would be meaningful. Hilbert-space fragmentation [29] embodying exponentially many disconnected Krylov subspaces is predicted to evade the strong version of ETH. Whereas its deterministic demonstration by scalable experiments is still pending [207,208]. Moreover, the connections between the ergodicity breaking and topological nature could be an intriguing topic for theory and experiments [26].

The alliance of ergodicity breaking and Floquet driving opens up avenues to explore new nonequilibrium systems. In addition to the DTC aforementioned, Stark many-body localization is also expected to enable the stability of DTC [200,209] with the aid of a sufficiently strong linear potential rather than strong disorders. Inspired by the recently effective generation of 60-qubit GHZ state enforced by the cat scar DTC [59], it is possible to further enlarge the scale of entangled states, associated with quantum many-body scar DTC, perthermal DTC, Stark many-body localization DTC, and others. More practically, scalable and high-precision manipulation of many-body nonequilibrium states on superconducting quantum processors enables many possible applications, such as ultra-sensitive sensing, quantum teleportation, and quantum metrology.

## Figures and Tables

**Figure 1 entropy-26-00592-f001:**
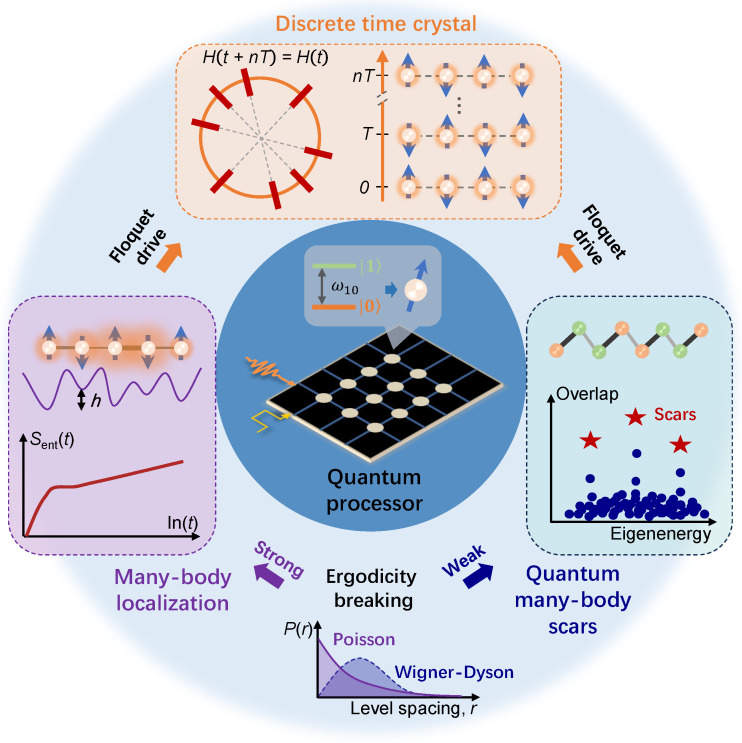
Superconducting quantum simulations of the nonequilibrium many-body physics. Quantum processor captured within the blue-circle regime, such as superconducting quantum processors, showcases the powerful capabilities in the simulation of many-body physics systems, thereof including many-body localization, quantum many-body scars, and discrete time crystal. The insert is the quantum processor with a two-level system mimicking the artificial atom. MBL can be generated in the interacting many-body system combined with strong disorder fluctuations (purple curve with a disorder strength of *h*), which uniquely owns the logarithmic growth in time of entanglement entropy, i.e., $S_{\mathrm{ent}}\left(t\right)\propto \ln\left(t\right)$. QMBS refers to several special eigenstates in the middle of the spectrum weakly violating ETH, where the spaced scars in towers could contribute to the coherent revivals of local observables. The common signature of MBL and QMBS is the ergodicity breaking. But the distinction is that MBL satisfies the Poisson distribution of energy level statistics (P(r)) which signals strong ergodicity breaking, whereas QMBS obeys the Wigner–Dyson distribution for weak breaking of ergodicity. Both MBL and QMBS can be exploited to protect DTC from being heated by Floquet driving, where the long-lived subharmonic response of local observables breaks the time-transition symmetry, i.e., $H\left(t+nT\right)=H\left(t\right),1<n\in N^{+}$.

**Figure 2 entropy-26-00592-f002:**
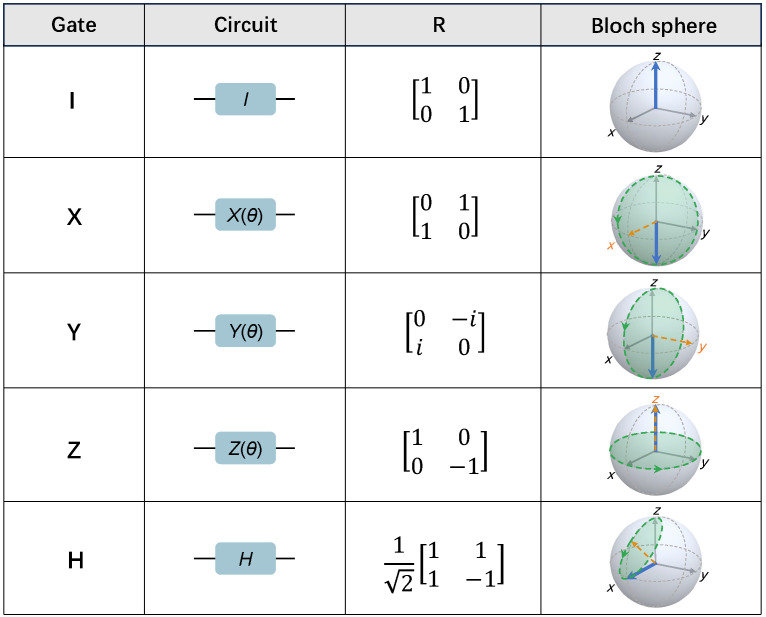
Description of single-qubit gates: I, X, Y, Z, and H gate. The gate name, circuit representation, rotation matrix *R* with rotation angle θ=π, and Bloch sphere representation of each gate are presented. The circuit representation will be utilized in the following context. The rotation matrix is based on eigenvectors of the $\sigma^{z}$ operator. Bloch sphere representations capture the corresponding quantum state (blue arrow) and its rotation around the axes (orange-dashed arrow) after operator R(θ=π) applied to the initial state of ∣↑〉 along *z* axis, where the rotation trajectory of the Bloch vector is unveiled by the green-dashed circuit.

**Figure 3 entropy-26-00592-f003:**
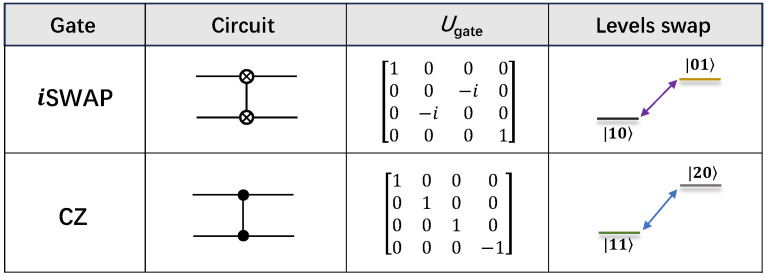
Canonical two-qubit gates: *i*SWAP and CZ gate. The gate name, circuit representation, process matrix *U*, and swap between the levels structure of both gates are illustrated.

**Figure 4 entropy-26-00592-f004:**
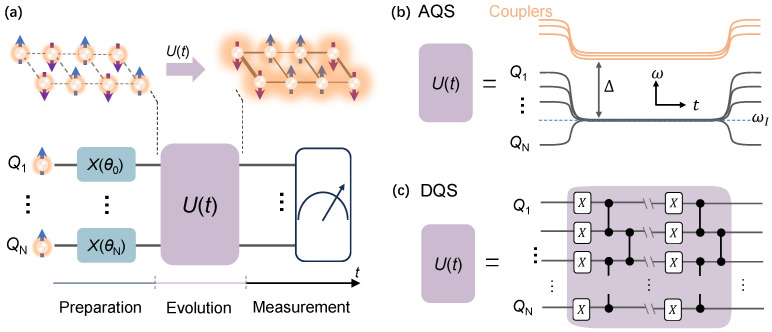
Schematic for many-body physics simulations. (**a**) The top panel describes the unitary evolution (*U*) of a quantum system from the initial state into the final state. The bottom panel depicts the three steps ordered in time (*t*) for quantum simulation: preparation, evolution, and measurement. The initial state is prepared via applying the XY rotation gates, such as $X(\theta_{i})$, to the corresponding qubit $Q_{i}$ in the ground state. After that, the system undergoes unitary evolution U(t) as the simulated physical model, where interactions and entanglement between qubits could be generated via an associated coupler (not indicated in the figure). Finally, as the unitary evolution finishes, states can be measured on demand, as indicated by the meter-like box. (**b**) AQS for the unitary evolution U(t). U(t) is implemented directly by engineering the couplings between qubits at the resonant frequency ($\omega_{I}$). The couplings is mediated by the frequency detuning (Δ) between couplers and interacting qubits. (**c**) DQS for the unitary evolution U(t). U(t) is decomposed into multilayer of logical gates operations.

**Figure 5 entropy-26-00592-f005:**
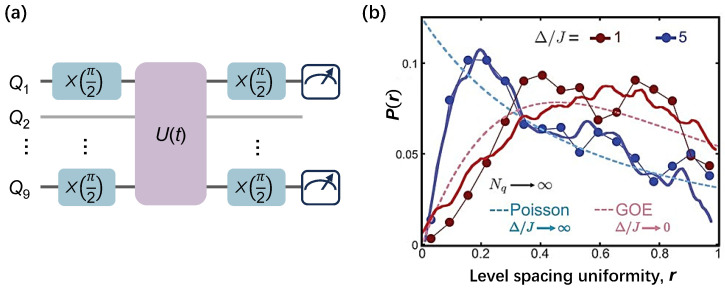
The spectroscopy statistical measurement via the many-body Ramsey spectroscopy technique. (**a**) Quantum circuits of the many-body Ramsey spectroscopy technique in a 9-qubit chain, where Ramsey pulses, such as single-qubit gate $X(\theta =\frac{\pi }{2})$ or $Y(\theta =\frac{\pi }{2})$, are applied to $Q_{1}$ and $Q_{9}$ simultaneously after the unitary evolution (U(t)) and before measurement. Here the system Hamiltonian H(t) is accomplished by tuning all qubits in resonance of the frequency domain with targeted couplings. It aims to place $Q_{1}$ and $Q_{9}$ in the superposition of |0〉 and |1〉 states and obtain the two-point corrections between them, and then construct $\chi_{2}\left(1,9\right)$. (**b**) The experimental spectroscopy statistical results for the 9-qubit 1D Bose–Hubbard model with disorder strength Δ/J=1 (red dotted line) and 5 (blue dotted line) in the case of two-photon interactions. The GOE (red dashed line) and Poisson type (blue dashed-line) distributions indicate the ergodic and nonergodic behavior of the quantum state, respectively. Panel (**b**) is adapted from [77].

**Figure 6 entropy-26-00592-f006:**
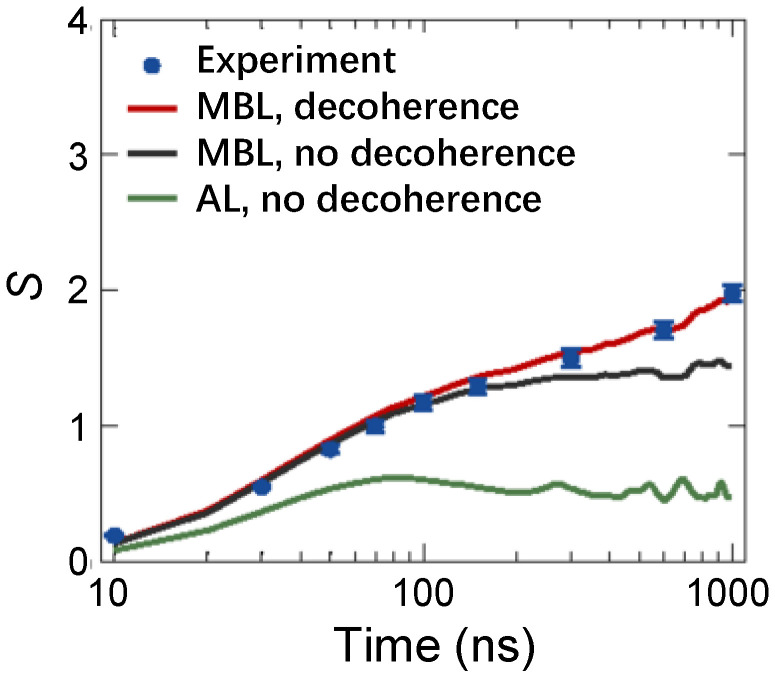
In a 10-qubit superconducting chain with long-range interactions, half-chain EE (*S*) was measured via QST technology. Time-dependent *S* are compared between MBL and AL. Wherein the lines infer to corresponding numerical simulation. The presence (absence) of decoherence is related to taking (not taking) long-range spin-spin interactions into account. Figures are adapted from [78].

**Figure 7 entropy-26-00592-f007:**
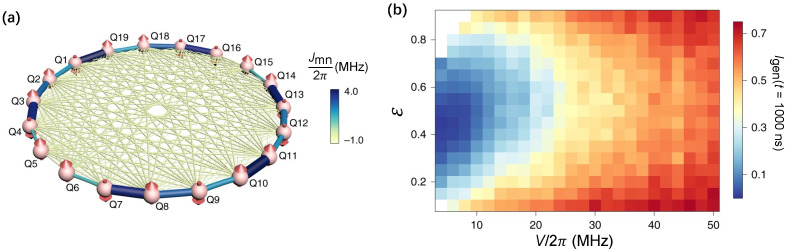
Energy-resolved many-body localization. (**a**) Diagram represents the 19 superconducting qubits with spin-spin interacting $J_{mn}/2\pi$ in an all-to-all layout, where the thickness and colors of bounds denote the coupling strength synchronously. (**b**) Energy-resolved (ε) and disorder (V/2π) averaged imbalance $I_{\mathrm{gen}}\left(t=1000\;\mathrm{ns}\right)$ in the experiment, which agrees well with the numerical simulations. Figures are adapted from [137].

**Figure 8 entropy-26-00592-f008:**
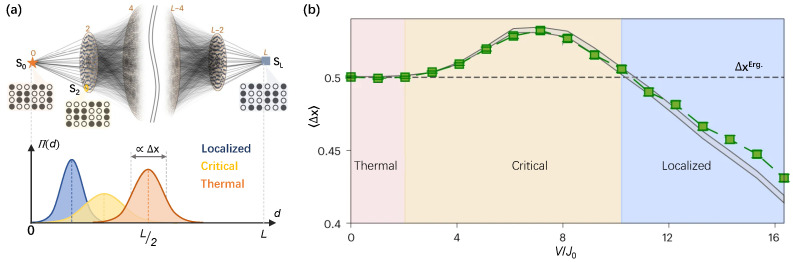
Observation of many-body dynamics from Fock space. (**a**) Top panel: the schematic illustration of network structure of the Hamming distance (*d*) distributed Fock state (s) of the 24-qubit system with half excitations, where the quantum system is initially prepared as $s_{0}$ (orange star) with d=0 as an apex of the network and farthest state $s_{L}$ (blue square) is antithetic state of $s_{0}$ as another apex. The massive local connectivity (grey lines) unveils the possible hopping between two sites with a Hamming distance difference of 2 in the Fock space network. Bottom panel: with the radial probability distribution Π(d), three different states (thermal, critical, and localized) could be identified. The double-headed arrow indicates the normalized width (Δx) of the Π(d) of a realization. (**b**) The experimental results (green squares) of realization-averaged normalized width 〈Δx〉 of wave packets for the equilibrium steady states as a function of disorder strength $V/J_{0}$, which agrees well with the numerical simulations (gray area). The dashed horizontal line denotes the normalized width of the ergodic wave pocket $\Delta x^{\mathrm{Erg}.}$. Figures are adapted from [141].

**Figure 9 entropy-26-00592-f009:**
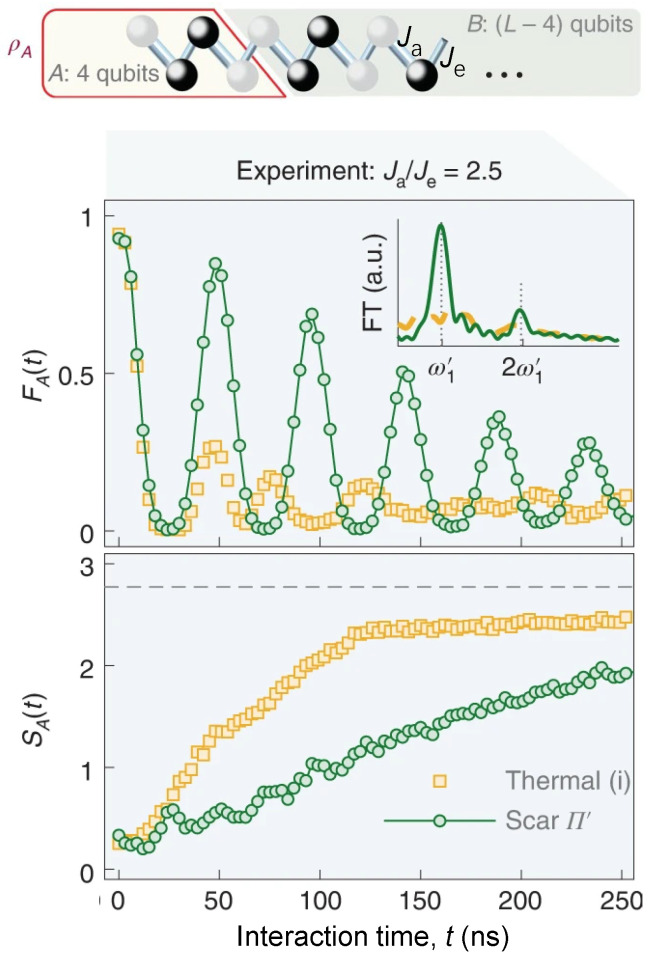
Demonstration of quantum many-body scars in a non-constrained chain. Top: Illustrative diagram shows the reduced density matrix $\rho_{A}$ of subregion A containing 4 qubits in a dimerized chain. $J_{a}$ and $J_{e}$ represent the intra-coupling and inter-coupling of the dimer formed by two adjacent qubits. $\rho_{A}$ can be utilized to compute the dynamical fidelity $F_{A}\left(t\right)=\mathrm{Tr}\left[\rho_{A}\left(0\right)\rho_{A}\left(t\right)\right]$ and the entanglement entropy $S_{A}\left(t\right)$ for various half-filling initial states. Bottom: $F_{A}\left(t\right)$ and $S_{A}\left(t\right)$ for the chain with couplings of $J_{a}/2\pi =2.5J_{e}/2\pi ≃-10\;\mathrm{MHz}$ for thermalizing initial state (orange squares) and quantum many-body scar state $|\Pi^{\prime }\rangle =|10011001\cdots \rangle$ or |01100110⋯〉 (green circles). Insert shows the Fourier transform of $F_{A}\left(t\right)$ with a highlight peak at $\omega_{1}^{\prime }$. Figures are adapted from [169].

**Figure 10 entropy-26-00592-f010:**
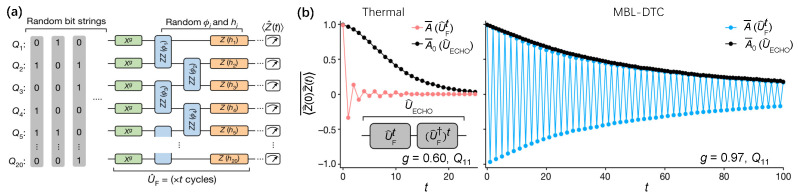
Observation of MBL-DTC. (**a**) Experimental circuits to generate the MBL-DTC with *t* identical cycles of unitary $U_{F}$. The random bit strings stand for the initial state of different disorder instances. (**b**) Dynamical disorder and initial-state averaged autocorrelators $\bar{A}=\bar{\langle Z(0)Z(t)\rangle }$ at qubit $Q_{11}$ are expressed distinctively for the thermal phase (red-dot line in left) and MBL-DTC phase (blue-dot line in right). Echo circuit $U_{\mathrm{ECHO}}={\left(U_{F}^{†}\right)}^{t}U_{F}^{t}$ reverses the Floquet evolution after *t*. Autocollerator $\bar{A_{0}}=\sqrt{\langle ZU_{\mathrm{ECHO}}^{†}ZU_{\mathrm{ECHO}}\rangle }$ quantifies the impact of decoherence. Figures are adapted from [54].

**Figure 11 entropy-26-00592-f011:**
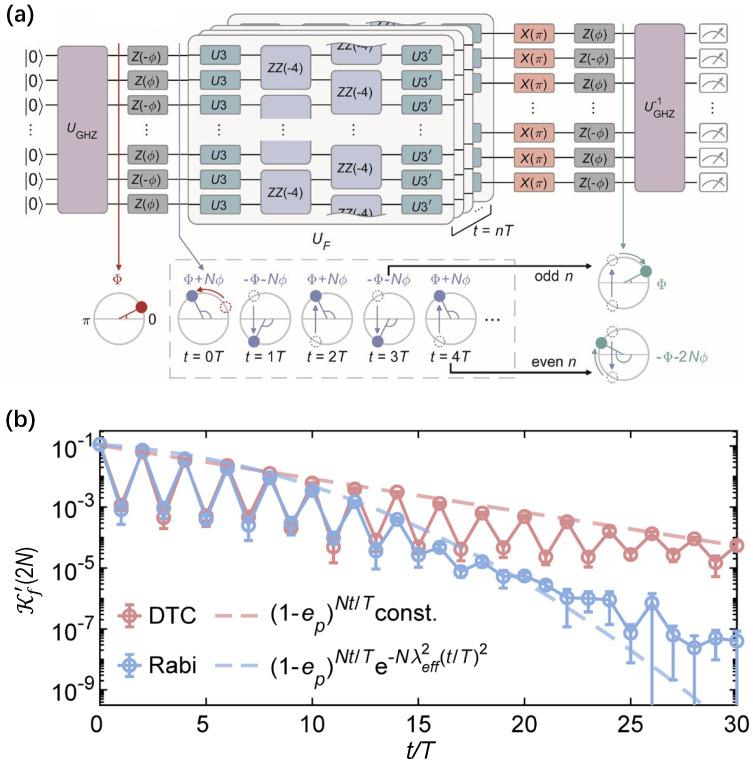
Application of the cat scar DTC. (**a**) Digital quantum circuits for cat scars interferometry, where the Floquet unitary $U_{F}$ is resolved into series single–qubit operations (U3 and $U3^{\prime }$) and two layers Ising interactions ZZ(−4). $U_{\mathrm{GHZ}}$ is unitary for preparing the cat scar states. (**b**) Measured ground state probability with Fourier transformation dynamics $K_{f}^{\prime }\left(2N\right)$ in cat scar DTC (red circles) and the contrast under noninteracting Rabi drives (blue circles). Corresponding dashed lines represent analytical results. Figures are adapted from [59].

## Data Availability

Not applicable.

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
