# Peer review of "Superconducting Quantum Simulation for Many-Body Physics beyond Equilibrium"

_entropy, 2024, doi:10.3390/e26070592_

Round 1

Reviewer 1 Report

Comments and Suggestions for Authors

This review presents significant experimental advancements in simulating nonequilibrium many-body systems with superconducting quantum circuits. Emphasizing the role of quantum superposition and entanglement, the review discusses how these systems circumvent thermalization, preserving local quantum data through mechanisms such as many-body localization (MBL) and quantum many-body scars (QMBS). Additionally, it highlights the development of discrete time crystals (DTCs), characterized by subharmonic motion due to disrupted time symmetry. Utilizing medium-scale quantum processors, these experiments surpass the limitations of classical approaches, offering new perspectives on complex many-body phenomena. The review is organized to provide foundational theories, detailed analyses of specific phenomena, and future prospects in superconducting quantum technologies. I will use this review myself to check references on the topic.

I would just add here a few refs from which the reader may benefit:

Digital-analog quantum simulations with superconducting circuits (Advances in Physics:X, 2017)
Circuit quantum electrodynamics (RMP,2021)
Quantum Electrodynamics in a Topological Waveguide (PRX,2021)
Stable quantum-correlated many-body states through engineered dissipation (Science,2024)
Dynamics of magnetization at infinite temperature in a Heisenberg spin chain (Science,2024)

Comments on the Quality of English Language

The review seems well written. I just realized one typo while reading, if there are more, I was not aware of them:

"An JJ" --> "A JJ" (Josephson junction)

Reviewer 2 Report

Comments and Suggestions for Authors

The manuscript briefly summarizes the recent developments of the quantum simulation experiments based on superconducting qubits. Quantum simulation is a central topic in modern physics, and can provide valuable insights to the non-equilibrium dynamics of many-body quantum physics. I can recommend this manuscript for publication in Entropy if the following comments are well addressed:

1. In the abstract, line 9, several phenomena (many-body localization, time crystal, etc) are mentioned. All of them are non-equilibrium phenomena. In line 8, after the word “exotic”, I recommend that the author should also add “non-equilibrium” to emphasize the topic of quantum simulation.

2. In the first paragraph of the introduction, it is important to emphasize the difficulty of classical computation, such as the exponential growth of the dimension of Hilbert space with increasing system sizes.

3. In the section 2.2, it is worthwhile to introduce how to estimate the fidelity of quantum gates and numerically simulate the digital quantum circuits with errors. The details should be provided.

4. In line 222, the Bose-Hubbard model is a hard-core version. The author could add “hard-core” before “Bose-Hubbard”.

5. In Fig. 8(b), the boundary between the “critical regime” and “localized regime” has been located. The authors should clarify the physical reason behind the estimation of the boundary. If the reason is NOT physically sound, I recommend that the author should replace Fig.8(b) with other plots in Ref.[127], such as Fig.S18(b) of the supplementary information.

6. The many-body localization in two-dimensional system is controversial. In line 348, “the existence of stable 2D MBL” can be replaced by “the delocalization in 2D disordered many-body quantum systems”.

7. In line 544, the sentence “The key challenge … in the critical regime” is misleading. I agree with the authors that finding an appropriate probe of many-body localization transition is an important task. However, the key challenge lies in the “nature” of the transition, which suffers from strong finite-size effect. One can overcome it by increasing the number of qubits, and provide experimental data with larger quantum simulators consisting of, such as, 100 qubits or 1000 qubits. This should be clarified in the section of outlook.

8. In lines 550 and 551, there is a discussion about the Hilbert-space fragmentation (HSF). Actually, for strong HSF, it strongly breaks the eigenstate thermalization hypothesis, while for weak HSF, it is a weak breakdown of ETH. Both the strong and weak HSF break the strong version of ETH. Therefore, in line 551, “weakly break the ergodicity” should be replaced by “break the strong version of ETH” or some descriptions like that.

Round 2

Reviewer 2 Report

Comments and Suggestions for Authors

I would like to thank the authors for the careful revision and reply. I recommend that this manuscript can be accepted by Entropy.